# Dibenzofuran, 4-Chromanone, Acetophenone, and Dithiecine Derivatives: Cytotoxic Constituents from *Eupatorium fortunei*

**DOI:** 10.3390/ijms22147448

**Published:** 2021-07-12

**Authors:** Chun-Hao Chang, Semon Wu, Kai-Cheng Hsu, Wei-Jan Huang, Jih-Jung Chen

**Affiliations:** 1Institute of Biopharmaceutical Sciences, School of Pharmaceutical Sciences, National Yang Ming Chiao Tung University, Taipei 112, Taiwan; changch.ps08@nycu.edu.tw; 2Department of Life Science, Chinese Culture University, Taipei 110, Taiwan; wsm4@ulive.pccu.edu.tw; 3Graduate Institute of Cancer Biology and Drug Discovery, College of Medical Science and Technology, Taipei Medical University, Taipei 110, Taiwan; piki@tmu.edu.tw; 4Ph.D. Program for Cancer Molecular Biology and Drug Discovery, College of Medical Science and Technology, Taipei Medical University, Taipei 110, Taiwan; 5Ph.D. Program in Biotechnology Research and Development, College of Pharmacy, Taipei Medical University, Taipei 110, Taiwan; wjhuang@tmu.edu.tw; 6Graduate Institute of Pharmacognosy, College of Pharmacy, Taipei Medical University, Taipei 110, Taiwan; 7Department of Pharmacy, School of Pharmaceutical Sciences, National Yang Ming Chiao Tung University, Taipei 112, Taiwan; 8Faculty of Pharmacy, National Yang-Ming University, Taipei 112, Taiwan; 9Department of Medical Research, China Medical University Hospital, China Medical University, Taichung 404, Taiwan

**Keywords:** *Eupatorium fortunei*, dibenzofuran, cytotoxity, apoptosis

## Abstract

Five new compounds, eupatodibenzofuran A (**1**), eupatodibenzofuran B (**2**), 6-acetyl-8-methoxy-2,2-dimethylchroman-4-one (**3**), eupatofortunone (**4**), and eupatodithiecine (**5**), have been isolated from the aerial part of *Eupatorium fortunei*, together with 11 known compounds (**6**‒**16**). Compounds **1** and **2** featured a new carbon skeleton with an unprecedented 1-(9-(4-methylphenyl)-6-methyldibe nzo[*b*,*d*]furan-2-yl)ethenone. Among the isolates, compound **1** exhibited potent inhibitory activity with IC_50_ values of 5.95 ± 0.89 and 5.55 ± 0.23 μM, respectively, against A549 and MCF-7 cells. The colony-formation assay demonstrated that compound **1** (5 μM) obviously decreased A549 and MCF-7 cell proliferation, and Western blot test confirmed that compound **1** markedly induced apoptosis of A549 and MCF-7 cells through mitochondrial- and caspase-3-dependent pathways.

## 1. Introduction

*Eupatorium fortunei* Turcz. (Asteraceae) is a perennial herb widely distributed in China. The aerial part of *E. fortunei* (Chinese name: Pei-Lan) has been used as a traditional medicine for the treatment of various diseases such as flu, poor appetite, constipation, nausea, and siriasis [1]. Diverse monoterpenoids [2,3,4,5,6], triterpenoids [2,7], sesquiterpenoids, stereoisomer, coumarins [7], pyrrolizidine alkaloids [7,8], benzofuran [4,9], and their derivatives were isolated from this species in the past studies. Many of these isolated compounds showed anti-inflammatory [5], anti-bacterial [3,4], anti-cancer [6], and anti-diabetic [7] activities. In our study on the cytotoxic constituents of Chinese herbal medicines, many species have been screened for cytotoxic effect, and *E. fortunei* has been found to be an active species. Two new dibenzo[*b,d*]furan derivatives, eupatodibenzofuran A (**1**) and eupatodibenzofuran B (**2**), a new 4-chromanone, 6-acetyl-8-methoxy-2,2-dimethylchroman-4-one (**3**), a new acetophenone derivative, eupatofortunone (**4**), a new dithiecine derivative, eupatodithiecine (**5**), and eleven known compounds (**6**–**16**) have been isolated and confirmed from the aerial part of *E. fortunei*. This report describes the structural elucidation of five new compounds **1**–**5** and the inhibitory activities of **1** against non-small-cell lung cancer and breast cancer cells.

## 2. Results and Discussion

### 2.1. General

Separation of the EtOAc-soluble fraction of a MeOH extract of the aerial part of *E. fortunei* by silica gel chromatography and preparative thin-layer chromatography (TLC) afforded five new (**1**–**5**) and eleven known compounds (**6**–**16**) (Figure 1).

### 2.2. Structure Elucidation of the New Compounds

Compound **1** was obtained as colorless needles. Its molecular formula C_22_H_18_O_4_ was deduced from a sodium adduct ion at *m/z* 369.11041 [M + Na]^+^ (calcd 369.11028) in the HRESIMS spectrum. The presence of hydroxyl (3439 cm^−1^) and conjugated carbonyl (1645 cm^−1^) groups were revealed from the IR spectrum. Analysis of the ^1^H and ^13^C NMR data of **1** showed the signals for an acetyl group [δ_H_ 2.41 (3H, s, Ac-2); δ_C_ 26.2 (COMe), 203.9 (COMe)], a chelated hydroxyl group [δ_H_ 12.65 (1H, s, OH-3)], a methyl group [δ_H_ 2.62 (3H, s, Me-6); δ_C_ 15.1 (Me-6)], a 2-hydroxy-4-methylphenyl group [δ_H_ 2.44 (3H, s, Me-4′), 4.97 (1H, s, OH-2′), 6.93 (1H, br d, *J* = 7.5 Hz, H-5′), 6.96 (1H, br s, H-3′), 7.25 (1H, d, *J* = 7.5 Hz, H-6′); δ_C_ 21.4 (Me-4′), 116.1 (C-3′), 121.7 (C-5′), 122.4 (C-1′), 130.2 (C-6′), 140.4 (C-4′), 152.7 (C-2′)], two *ortho*-coupled aromatic protons [δ_H_ 7.22 (1H, d, *J* = 7.5 Hz, H-8), 7.33 (1H, br d, *J* = 7.5 Hz, H-7); δ_C_ 125.1 (C-8), 128.6 (C-7)], and two singlet aromatic protons [δ_H_ 7.07 (1H, s, H-4), 7.65 (1H, s, H-1); δ_C_ 99.8 (C-4), 125.4 (C-1)] (Table 1). The position of each substituent was supported by ROESY correlations between H-1 (δ_H_ 7.65)/Ac-2 (δ_H_ 2.41), Me-6 (δ_H_ 2.62)/H-7 (δ_H_ 7.33), OH-2′ (δ_H_ 4.97)/H-3′ (δ_H_ 6.96), H-3′ (δ_H_ 6.96)/Me-4′ (δ_H_ 2.44), and Me-4′ (δ_H_ 2.44)/H-5′ (δ_H_ 6.93) and by HMBC correlation between ‘Ac-2 (δ_H_ 2.41)/C-2 (δ_C_ 116.6)’, ‘OH-3 (δ_H_ 12.65)/C-2 (δ_C_ 116.6), C-4 (δ_C_ 99.8)’, ‘Me-6 (δ_H_ 2.62)/C-5a (δ_C_ 155.9), C-7 (δ_C_ 128.6)’, ‘OH-2′ (δ_H_ 4.97)/C-3′ (δ_C_ 116.1)’, and ‘Me-4′ (δ_H_ 2.44)/C-4′ (δ_C_ 140.4), C-5′ (δ_C_ 121.7)’ (Figure 2). The full assignment of ^1^H and ^13^C NMR resonances was supported by ^1^H–^1^H COSY, HSQC, ROESY (Figure 2A), and HMBC (Figure 2B) spectral analyses. Thus, the structure of **1** was elucidated as 1-(3-hydroxy-9-(2-hydroxy-4-methylphenyl)-6-methyldibenzo[*b*,*d*]furan-2-yl)ethenone, named eupatodibenzofuran A.

Compound **2** was isolated as amorphous powder with molecular formula C_23_H_20_O_4_, as determined by positive-ion HRESIMS, showing an [M + Na]^+^ ion at *m/z* 383.12612 (calcd for C_23_H_20_O_4_Na, 383.12593). The presence of a conjugated carbonyl group was revealed by a band at 1645 cm^−1^ in the IR spectrum, and was confirmed by the resonance at δ_C_ 203.7 in the ^13^C NMR spectrum. The ^1^H NMR data of **2** were similar to eupatodibenzofuran A (**1**), except that the 2′-methoxy group [δ_H_ 3.70 (3H, s), δ_C_ 55.5] of **2** replaced the 2′-hydroxy group [δ_H_ 4.97 (1H, s)] of **1**. This was supported by the ROESY correlations between OMe-2′ (δ_H_ 3.70) and H-3′ (δ_H_ 6.91) and by the HMBC correlations between OMe-2′ (δ_H_ 3.70) and C-2′ (δ_C_ 156.7) (Table 1). The position of each substituent was supported by ROESY correlations between H-1 (δ_H_ 7.59)/Ac-2 (δ_H_ 2.40), Me-6 (δ_H_ 2.60)/H-7 (δ_H_ 7.27), H-7 (δ_H_ 7.27)/H-8 (δ_H_ 7.17), OMe-2′ (δ_H_ 3.70)/H-3′ (δ_H_ 6.91), H-3′ (δ_H_ 6.91)/Me-4′ (δ_H_ 2.49), and Me-4′ (δ_H_ 2.49)/H-5′ (δ_H_ 6.95) and by HMBC correlation between ‘H-1 (δ_H_ 7.59)/MeCO-2 (δ_C_ 203.7), C-3 (δ_C_ 162.9), C-4a (δ_C_ 161.3), C-9a (δ_C_ 121.7)’, ‘Ac-2 (δ_H_ 2.40)/C-2 (δ_C_ 116.0)’, ‘OH-3 (δ_H_ 12.62)/C-2 (δ_C_ 116.0)’, ‘H-4 (δ_H_ 7.05)/C-2 (δ_C_ 116.0), C-9b (δ_C_ 117.8)’, ‘Me-6 (δ_H_ 2.60)/C-5a (δ_C_ 155.4), C-7 (δ_C_ 127.9)’, ‘H-7 (δ_H_ 7.27)/C-5a (δ_C_ 155.4)’, ‘H-8 (δ_H_ 7.17)/C-6 (δ_C_ 120.5), C-9a (δ_C_ 121.7), C-1′ (δ_C_ 125.4)’, ‘H-6′ (δ_H_ 7.28)/C-9 (δ_C_ 130.6), C-2′ (δ_C_ 156.7), C-4′ (δ_C_ 139.8)’, ‘OMe-2′ (δ_H_ 3.70)/C-2′ (δ_C_ 156.7)’, ‘Me-4′ (δ_H_ 2.49)/C-4′ (δ_C_ 139.8), C-5′ (δ_C_ 121.4)’ (Figure 3). The full assignment of ^1^H and ^13^C NMR resonances was confirmed by ^1^H–^1^H COSY, ROESY (Figure 3A), HSQC, and HMBC (Figure 3B) techniques. According to the evidence above, the structure of **2** was elucidated as 1-(3-hydroxy-9-(2-methoxy-4-methylphenyl)-6-methyldibenzo[*b*,*d*]furan-2-yl)ethan-1-one, named eupatodibenzofuran B.

Compound **3** was isolated as light brown amorphous powder. Its molecular formula, C_14_H_16_O_4_, was determined on the basis of the positive HRESIMS at *m/z* 249.1123 [M + H]^+^ (calcd 249.1121) and was supported by the ^1^H and ^13^C NMR data. The IR absorption bands of **3** revealed the presence of carbonyl (1686 cm^−1^) function. Analyses of the ^1^H and ^13^C NMR data of **3** showed the signals for two methyl groups [δ_H_ 1.55 (6H, s, Me-2 × 2); δ_C_ 26.5 (Me-2 × 2)], an acetyl group [δ_H_ 2.60 (3H, s, Ac-6); δ_C_ 26.2 (COMe), 196.6 (COMe)], a methoxy group [δ_H_ 3.95 (3H, s, OMe-8); δ_C_ 56.4 (OMe-8)], two *meta*-coupled aromatic protons [δ_H_ 7.70, 8.08 (each 1H, each d, *J* = 2.0 Hz, H-7 and H-5); δ_C_ 114.3 (C-7), 120.0 (C-5)], and two methylene protons [δ_H_ 2.79 (2H, s, H-3); δ_C_ 48.5 (C-3)] (Table 2). The position of each substituent was supported by the ROESY correlations between Me-2 (δ_H_ 1.55)/H-3 (δ_H_ 2.79), H-5 (δ_H_ 8.08)/Ac-6 (δ_H_ 2.60), Ac-6 (δ_H_ 2.60)/H-7 (δ_H_ 7.70), and H-7 (δ_H_ 7.70)/OMe-8 (δ_H_ 3.95) and HMBC correlations between ‘Me-2 (δ_H_ 1.55)/C-2 (δ_C_ 81.0), C-3 (δ_C_ 48.5)’, ‘H-3 (δ_H_ 2.79)/C-2 (δ_C_ 81.0), C-4 (δ_C_ 191.7), Me-2 (δ_C_ 26.5)’, ‘H-5 (δ_H_ 8.08)/C-4 (δ_C_ 191.7), MeCO-6 (δ_C_ 196.6), C-7 (δ_C_ 114.3), C-8a (δ_C_ 154.0)’, ‘H-7 (δ_H_ 7.70)/C-5 (δ_C_ 120.0), MeCO-6 (δ_C_ 196.6), C-8a (δ_C_ 154.0)’, and ‘OMe-8 (δ_H_ 3.95)/C-8 (δ_C_ 149.8)’ (Figure 4). The full assignment of ^1^H and ^13^C NMR resonances was further confirmed by the ^1^H-^1^H COSY, HSQC, 1D-selective NOESY (Figure 4A), and HMBC (Figure 4B) data. Consequently, the structure of compound **3** was established as 6-acetyl-8-methoxy-2,2-dimethylchroman-4-one.

Compound **4** was obtained as colorless oil and the molecular formula was determined to be C_14_H_16_O_3_ by HRESIMS [*m/z* 255.10030 [M + Na]^+^ (calcd for C_14_H_16_O_3_Na, 255.09971)]. The IR spectrum showed the presence of ester and conjugated carbonyl groups at 1736 and 1684 cm^−1^, respectively. Analysis of the ^1^H and ^13^C NMR data of **4** revealed the signals for an acetyl group [δ_H_ 2.52 (3H, s, Ac-2); δ_C_ 29.4 (COMe), 197.1 (COMe)], a methyl group [δ_H_ 2.40 (3H, s, Me-5); δ_C_ 21.4 (Me-5)], a (*Z*)-(2-methylbut-2-enoyl)oxy group [δ_H_ 2.08 (3H, dq, *J* = 7.3, 1.3 Hz, H-4′), 2.08 (3H, qd, *J* = 1.8, 1.3 Hz, H-5′), 6.30 (1H, qq, *J* = 7.3, 1.8 Hz, H-3′); δ_C_ 16.0 (C-4′), 20.6 (C-5′), 127.0 (C-2′), 141.4 (C-3′), 166.1 (C-1′)], and three mutually coupled aromatic protons [δ_H_ 6.94 (1H, br s, H-6), 7.12 (1H, br d, *J* = 8.0 Hz, H-4), and 7.74 (1H, d, *J* = 8.0 Hz, H-3); δ_C_ 124.4 (C-6), 126.7 (C-4), and 130.4 (C-3)] (Table 3). The position of each substituent was supported by the ROESY correlations between Ac-2 (δ_H_ 2.52)/H-3 (δ_H_ 7.74), H-3 (δ_H_ 7.74)/H-4 (δ_H_ 7.12), H-4 (δ_H_ 7.12)/Me-5 (δ_H_ 2.40), Me-5 (δ_H_ 2.40)/H-6 (δ_H_ 6.94), H-3′ (δ_H_ 6.30)/H-4′ (δ_H_ 2.08), and H-3′ (δ_H_ 6.30)/H-5′ (δ_H_ 2.08) and by HMBC correlations between ‘COMe-2 (δ_H_ 2.52)/C-2 (δ_C_ 128.3), COMe-2 (δ_C_ 197.1)’, ‘H-3 (δ_H_ 7.74)/C-1 (δ_C_ 149.3), COMe-2 (δ_C_ 197.1), C-5 (δ_C_ 144.7)’, ‘H-4 (δ_C_ 7.12)/C-2 (δ_C_ 128.3), C-6 (δ_C_ 124.4), Me-5 (δ_C_ 21.4)’, ‘Me-5 (δ_H_ 2.40)/C-4 (δ_C_ 126.7), C-5 (δ_C_ 144.7), C-6 (δ_C_ 124.4)’, ‘H-6 (δ_H_ 6.94)/C-1 (δ_C_ 149.3), C-2 (δ_C_ 128.3), C-4 (δ_C_ 126.7), Me-5 (δ_C_ 21.4)’, ‘H-4′ (δ_H_ 2.08)/C-2′ (δ_C_ 127.0)’, and ‘H-5′ (δ_H_ 2.08)/C-1′ (δ_C_ 166.1), C-2′ (δ_C_ 127.0), C-3′ (δ_C_ 141.4)’ (Figure 5). The full assignment of ^1^H and ^13^C NMR resonances was supported by ^1^H–^1^H COSY, HSQC, ROESY (Figure 5A), and HMBC (Figure 5B) spectral analyses. According to the above data, the structure of **4** was elucidated as (*Z*)-2-acetyl-5-methylphenyl 2-methylbut-2-enoate, named eupatofortunone.

Compound **5** was isolated as light brown amorphous powder. Its molecular formula, C_20_H_20_O_4_S_2_, was determined on the basis of the positive HRESIMS at *m/z* 411.07026 [M + Na]^+^ (calcd 411.07007) and was supported by the ^1^H and ^13^C NMR data. The IR absorption bands of **5** revealed the presence of alkynyl (2230 cm^−1^) and conjugated carbonyl (1639 cm^−1^) functions. Analysis of the ^1^H and ^13^C NMR data of **5** revealed the signals for an acetyl group [δ_H_ 2.50 (3H, s, Ac-2); δ_C_ 29.4 (COMe-2), 190.2 (COMe-2)], a prop-1-yn-1-yl group [δ_H_ 2.09 (3H, s, C≡CMe); δ_C_ 4.8 (C≡CMe), 73.2 (C≡CMe), 94.8 (C≡CMe)], a methoxy group [δ_H_ 3.94 (3H, s, OMe-5); δ_C_ 58.8 (OMe-5)], and a singlet aromatic proton [δ_H_ 6.84 (1H, s, H-4); δ_C_ 119.1 (C-4)] (Table 4). The position of each substituent was supported by the HMBC correlation between ‘COMe (δ_H_ 2.50)/C-2 (δ_C_ 122.6), COMe-2 (δ_C_ 190.2)’, ‘C≡CMe (δ_H_ 2.09)/C≡CMe (δ_C_ 73.2), C≡CMe (δ_C_ 94.8)’, ‘H-4 (δ_H_ 6.84)/C-2 (δ_C_ 122.6), C-3 (δ_C_ 130.6), C≡CMe (δ_C_ 73.2)’, ‘OMe-5 (δ_H_ 3.94)/C-5 (δ_C_ 159.3)’ and by the 1D selective NOESY correlation between H-4 (δ_H_ 6.84) and OMe-5 (δ_H_ 3.94) (Figure 6). According to the ^1^H, ^13^C NMR, and HR-ESI-MS data, the number of resonances observed was half that expected, suggesting that **5** had a symmetrical structure. The full assignment of ^1^H and ^13^C NMR resonances was further confirmed by ^1^H-^1^H COSY, 1D-selective NOESY (Figure 6A), HSQC, and HMBC (Figure 6B) data. Consequently, the structure of compound **5** was established as 1,1′-((*2E*,*4Z*,*7Z*,*9E*)-5,7-dimethoxy-3,9-di(prop-1-yn-1-yl)-1,6-dithiecine-2,10-diyl)diethanone, named eupatodithiecine.

### 2.3. Structure Identification of the Known Isolated Compounds

The known compounds were readily identified by a comparison of their physical and spectroscopic data (UV, IR, ^1^H NMR, and MS) with those of authentic samples or literature values. They include four thymol derivatives, thymyl angelate (**6**) [10], 8,9-dehydrothymol 3-*O*-tiglate (**7**) [4], 9-angeloyloxythymol (**8**) [5], and 9-*O*-angeloyl-8,10-dehydrothymol (**9**) [11], five phenol derivatives, 2-hydroxy-4-methylacetophenone (**10**) [12], *trans*-*o*-coumaric acid (**11**) [13], 6-hydroxy-7-methoxy-2-isopropenyl-5-acetylcumaran (**12**) [14], 2,4-di-*tert*-butylphenol (**13**) [15], and 1-(2-hydroxy-5-methoxy-4-methylphenyl)ethanone (**14**) [16], a coumarin (**15**) [17], and a triterpenoid, taraxasterol (**16**) [18].

### 2.4. Biological Studies

The cytotoxic effects of the isolated compounds from *E. fortunei* were evaluated by their activities to suppress A549 and MCF-7 cells. The cytotoxic activity data are shown in Table 5. Among the isolated compounds, compound **1** exhibited potent inhibitory activities with IC_50_ values of 5.95 ± 0.89 and 5.32 ± 0.31 μM, respectively, against A549 and MCF-7 cells. In addition, colony-formation assay was performed to estimate the effects of compound **1** on the proliferation of A549 and MCF-7 cells. As shown in Figure 7, compound **1** reduced colony formation in a dose-dependent manner in both cell lines. In addition, we generated physicochemical properties (Appendix A) of compound **1** using Pipeline Pilot [19]. The ALogP98 value, molecular polar surface area, and ADMET absorption level suggest that compound **1** is hydrophobic and may have good permeability to cross the cell membrane, which may account for the cytotoxic effects of compound **1**.

To further confirm whether apoptosis was triggered, annexin V/propidium iodide (PI) assay was performed and the expression levels of apoptosis-associated proteins, Bcl-2, Bax, and caspase-3 were analyzed by Western blot analysis after A549 and MCF-7 cells were treated with compound **1**. As shown in Figure 8A,B, compound **1** significantly induced the cell apoptosis in A549 and MCF-7 cells, respectively. Furthermore, compound **1** increased the expression of Bax and cleaved-caspase-3, and decreased Bcl-2 and pro-caspase-3 levels in a dose-dependent manner in both A549 and MCF-7 cells (Figure 9A,B). The above results confirm that compound **1** markedly induces apoptosis of A549 and MCF-7 cells through mitochondrial- and caspase-3-dependent pathways (Scheme 1).

To further understand the mechanism of compound **1** in this study, we predicted potential targets using the similarity ensemble approach server [20]. This approach predicts possible target proteins of a compound by comparing chemical similarities. Compound **1** was predicted to target four proteins (Appendix A), including PON1, CELA1, CBR1, and NQO1. The Tanimoto coefficients (Tc) of chemical similarity were generated for the predicted targets. The Tc is a pairwise score between the compound and the predicted target. The Tc score ranges from 0.0 (no similarity) to 1.0 (total similarity). The P-value indicates the prediction reliability, and a value approaching zero means a reliable prediction. The possible targets may account for the inhibition mechanisms of compound **1**.

## 3. Materials and Methods

### 3.1. General Experimental Procedures

Ultraviolet (UV) spectra were obtained on a Jasco UV-240 spectrophotometer. Infrared (IR) spectra (neat or KBr) were recorded on a Perkin Elmer 2000 FT-IR spectrometer.

Nuclear magnetic resonance (NMR) spectra, including correlation spectroscopy (COSY), rotating frame nuclear Overhauser effect spectroscopy (ROESY), nuclear Overhauser effect spectroscopy (NOESY), heteronuclear multiple-bond correlation (HMBC), and heteronuclear single-quantum coherence (HSQC) experiments, were acquired using a BRUKER AVIII-500 spectrometer (Bruker, Bremen, Germany), operating at 500 MHz (^1^H) and 125 MHz (^13^C), respectively, with chemical shifts given in the ppm (δ) using tetramethylsilane (TMS) as an internal standard. Electrospray ionization (ESI) and high-resolution electrospray ionization (HRESI)-mass spectra were recorded on a Bruker APEX II Mass Spectrometer (Bruker, Bremen, Germany). Silica gel [70–230 mesh (63–200 μm) and 230–400 mesh (40–63 μm), Merck] was used for column chromatography (CC). Silica gel 60 F-254 (Merck, Darmstadt, Germany) was used for thin-layer chromatography (TLC) and preparative thin-layer chromatography (PTLC).

### 3.2. Plant Material

The aerial part of *E. fortunei* collected from Dihua St., Datong Dist., Taipei City, Taiwan, in May 2019 and identified by Prof. J.-J. Chen. A voucher specimen was deposited in the Department of Pharmacy, National Yang Ming Chiao Tung University, Taipei, Taiwan.

### 3.3. Extraction and Isolation

The aerial part of *E. fortunei* (5.0 kg) was pulverized and extracted three times with MeOH (30 L each) for 3 days. The MeOH extract was concentrated under reduced pressure at 35 °C, and the residue (123.7 g) was partitioned between EtOAc and H_2_O (1:1) to provide the EtOAc-soluble fraction (fraction A, 25.6 g). Fraction A (25.6 g) was chromatographed on silica gel (70–230 mesh, 1.3 kg), eluting with *n*-hexane, gradually increasing the polarity with EtOAc to give ten fractions: A1 (1 L, *n*-hexane/EtOAc, 100:1), A2 (1 L, *n*-hexane/EtOAc, 80:1), A3 (1.5 L, *n*-hexane/EtOAc, 70:1), A4 (2 L, *n*-hexane/EtOAc, 50:1), A5 (2 L, *n*-hexane/EtOAc, 30:1), A6 (2 L, *n*-hexane/EtOAc, 10:1), A7 (3.5 L, *n*-hexane/EtOAc, 5:1), A8 (2.5 L, *n*-hexane/EtOAc, 3:1), A9 (3 L, *n*-hexane/EtOAc, 1:1), and A10 (2.5 L, EtOAc). Fraction A2 (1.8 g) was subjected to column chromatography (CC) (85 g of silica gel, 230–400 mesh, *n*-hexane/acetone, 50:1–0:1, 250 mL–fractions) to give 11 subfractions: A2-1–A2-11. Part (125 mg) of fraction A2-2 was further purified by preparative TLC (silica gel, *n*-hexane/EtOAc, 30:1) to yield thymyl angelate (**6**) (5.2 mg) and 8,9-dehydrothymol 3-*O*-tiglate (**7**) (4.1 mg). Part (95 mg) of fraction A2-5 was further purified by preparative TLC (silica gel, *n*-hexane/CH_2_Cl_2_, 9:1) to obtain 2-hydroxy-4-methylacetophenone (**10**) (3.2 mg). Part (76 mg) of fraction A2-8 was further purified by preparative TLC (silica gel, *n*-hexane/EtOAc, 19:1) to afford eupatofortunone (**4**) (4.6 mg). Fraction A3 (1.6 g) was subjected to CC (72 g of silica gel, 230–400 mesh, *n*-hexane/EtOAc, 20:1–0:1, 250 mL–fractions) to give ten subfractions: A3-1–A3-10. Part (135 mg) of fraction A3-9 was further purified by preparative TLC (silica gel, *n*-hexane/acetone, 5:1) to obtain coumarin (**15**) (7.8 mg). Fraction A4 (2.3 g) was purified by medium pressure liquid chromatography (MPLC) (105 g of silica gel, 230–400 mesh, *n*-hexane/acetone, 19:1–0:1, 250 mL–fractions) to give eight subfractions: A4-1–A4-8. Fraction A4-5 (265 mg) was purified by MPLC (11.9 g of silica gel, *n*-hexane/EtOAc, 7:1) to afford four subfractions (each 150 mL, A4-5-1–A4-5-4). Fraction A4-5-3 (48 mg) was further purified by preparative TLC (silica gel, CH_2_Cl_2_/EtOAc, 9:1) to obtain *trans*-*O*-coumaric acid (**11**) (8.9 mg). Fraction A4-7 (53 mg) was further purified by preparative TLC (silica gel, CH_2_Cl_2_/acetone, 8:1) to obtain 2,4-di-*tert*-butylphenol (**13**) (3.2 mg). Faction A6 (2.2 g) was subjected to MPLC (100 g of silica gel, 230–400 mesh; CH_2_Cl_2_/EtOAc, 15:1–0:1, 250 mL–fractions) to give 13 subfractions: A6-1–A6-13. Part (92 mg) of fraction A6-6 was further purified by preparative TLC (silica gel, CH_2_Cl_2_/acetone, 7:1) to afford eupatodibenzofuran A (**1**) (4.3 mg) and eupatodibenzofuran B (**2**) (3.2 mg). Fraction A6-7 (226 mg) was purified by MPLC (silica gel, CH_2_Cl_2_/acetone, 4:1) to afford 5 subfractions (A6-7-1–A6-7-5, each 150 mL). Fraction A6-7-2 (49 mg) was further purified by preparative TLC (silica gel, CH_2_Cl_2_/EtOAc, 5:1) to obtain taraxasterol (**16**) (4.7 mg). Part (92 mg) of fraction A6-10 was further purified by preparative TLC (silica gel, CH_2_Cl_2_/EtOAc, 4:1) to yield 9-angeloyloxythymol (**8**) (5.1 mg) and 9-*O*-angeloyl-8,10-dehydrothymol (**9**) (4.3 mg). Part (111 mg) of fraction A6-11 was purified by preparative TLC (silica gel, CH_2_Cl_2_/EtOAc, 2:1) to obtain 6-hydroxy-7-methoxy-2-isopropenyl-5-acetylcumaran (**12**) (5.3 mg). Part (75 mg) of fraction A6-13 was further purified by semipreparative normal-phase high-performance liquid chromatography (HPLC) (silica gel, CH_2_Cl_2_/EtOAc, 4:1, 2.0 mL min^−1^) to afford 1-(2-hydroxy-5-methoxy-4-methylphenyl)ethanone (**14**) (4.2 mg). Faction A8 (1.6 g) was subjected to CC (72 g of silica gel, 230–400 mesh, CH_2_Cl_2_/MeOH 9:1–0:1, 250 mL–fractions) to give ten subfractions: A8-1–A8-10. Fraction A8-4 (88 mg) was further purified by semipreparative normal-phase HPLC (silica gel, CH_2_Cl_2_/MeOH, 9:1, 2.0 mL min^−1^) to afford eupatodithiecine (**5**) (3.7 mg). Part (93 mg) of fraction A8-7 was further purified by preparative TLC (silica gel, CH_2_Cl_2_/MeOH, 4:1) to yield 6-acetyl-8-methoxy-2,2-dimethylchroman-4-one (**3**) (4.4 mg).

Eupatodibenzofuran A (**1**): Colorless needles; mp 153–155 °C (CH_2_Cl_2_); UV (MeOH) λ_max_ (log ε) 238 (4.09), 272 (3.97), 352 (1.43) nm; IR (KBr) υ_max_ 3439 (OH), 1645 (C=O) cm^−1^; ^1^H and ^13^C NMR data, see Table 1; HRESIMS *m/z* 369.11041 [M + Na]^+^ (calcd for C_22_H_18_O_4_Na, 369.11028); HRESIMS, 1D-, and 2D-NMR spectra, see Appendix A.

Eupatodibenzofuran B (**2**): Amorphous powder; UV (MeOH) λ_max_ (log ε) 238 (4.29), 271 (4.12), 348 (3.24) nm; IR (neat) υ_max_ 3429 (OH), 1645 (C=O) cm^−1^; ^1^H and ^13^C NMR data, see Table 1; HRESIMS *m/z* 383.12612 [M + Na]^+^ (calcd for C_23_H_20_O_4_Na, 383.12593); HRESIMS, 1D-, and 2D-NMR spectra, see Appendix A.

6-Acetyl-8-methoxy-2,2-dimethylchroman-4-one (**3**): Light brown amorphous powder; UV (MeOH) λ_max_ (log ε) 247 (4.08), 280 (sh, 3.64), 329 (3.24) nm; IR (neat) υ_max_ 1686 (C=O) cm^−1^; ^1^H and ^13^C NMR data, see Table 2; HRESIMS *m/z* 249.1123 [M + H]^+^ (calcd for C_14_H_17_O_4_, 249.1121); HRESIMS, 1D-, and 2D-NMR spectra, see Appendix A.

Eupatofortunone (**4**): Colorless oil; UV (MeOH) λ_max_ (log ε) 211 (4.18), 246 (sh, 3.97), 284 (sh, 3.06) nm; IR (neat) υ_max_ 1736 (C=O), 1684 (C=O) cm^−1^; ^1^H and ^13^C NMR data, see Table 3; HRESIMS *m/z* 255.10030 [M + Na]^+^ (calcd for C_14_H_16_O_3_Na, 255.09971); HRESIMS, 1D-, and 2D-NMR spectra, see Appendix A.

Eupatodithiecine (**5**): Light brown amorphous powder; UV (MeOH) λ_max_ (log ε) 227 (4.03), 323 (4.26) nm; IR (neat) υ_max_ 2230 (C≡C), 1639 (C=O) cm^−1^; ^1^H and ^13^C NMR data, see Table 4; HRESIMS *m/z* 411.07026 [M + Na]^+^ (calcd for C_20_H_20_O_4_S_2_Na, 411.07007); HRESIMS, 1D-, and 2D-NMR spectra, see Appendix A.

### 3.4. Biological Assay

#### 3.4.1. Cell Culture

All cell lines were cultured at 37 °C under a humidified atmosphere with 5% CO_2_. Human non-small-cell lung cancer cell (A549) and human breast cancer cell (MCF-7) were obtained from American Type Culture Collection (ATCC, Rockville, MD, USA) and cultivated in Dulbecco’s modified Eagle’s medium (DMEM) (Himedia, India) supplemented with 10% heat-inactivated fetal bovine serum (FBS) (Avantor Seradigm/VWR, Radnor, PA, USA), 100 U/mL penicillin and 100 µg/mL streptomycin (Himedia, India). Cells were washed by warm phosphate buffered saline (PBS) every day and changed medium every 2–3 days [21,22].

#### 3.4.2. Cell Viability Assay

The cell viability was conducted by the MTT assay as previously described method [23]. Briefly, 5 × 10^3^ cells in 200 μL per well were plated in 96-well culture plates and cultured in complete medium overnight. After 24 h, cells were treated with different concentrations (3.125, 6.25, 12.5, 25, 50, and 100 μM) of compounds **1**–**16**. Fluorouracil (5-FU) (Sigma-Aldrich, St. Louis, MO, USA) was used as a positive control against A549 and MCF-7 cells with IC_50_ values of 10.57 ± 1.89 and 8.59 ± 1.03 μM, respectively. The optical density at 570 nm was measured by ELISA plate reader (μ Quant) and the IC_50_ value was calculated. The optical density of formazan formed in control (untreated) cells was taken as 100% viability.

#### 3.4.3. Colony-Formation Assay

The colony-formation assay was determined by the reference method with a slight modification [24]. In this assay, A549 and MCF-7 cells were seeded in 6-well plates with 1 × 10^3^ cells per well and incubated for 12 h. The cells were then treated with the indicated concentrations of compound **1**, and cultured for 10 days. The cells were washed three times using PBS and fixed using 95% methanol for 30 min. After washing three times with distilled water, the cells were stained using 0.2% crystal violet dye for 20 min and rinsed with distilled water to wash away the excess dye. The visible colonies were compared with the control samples and photographed using a standard camera under natural light.

#### 3.4.4. Flow Cytometry

Annexin V/PI assay was used to determine the apoptotic and necrotic cells. The A549 and MCF-7 cells were seeded on 6-well microplates at a density of 10^6^ cells/mL respectively. After 24 h incubation, the cells were treated with following concentrations of 0, 5, and 10 μM for compound **1**. After 24 h, they were washed and re-suspended in PBS solution (500 μL). Then, Annexin V-FITC (5 μL) and PI staining solution (5 μL) were introduced to the mixture, and the incubation process was followed under the dark condition for 5 min at 25 °C. Finally, flow cytometer analysis (Beckman Coulter^®^, Miami, FL, USA) was performed using an AnnexinV-FITC Apoptosis Detection Kit (Strong Biotech Corporation, Taipei, Taiwan) and Flowjo version 7.6.1. Software.

#### 3.4.5. Western Blot

Western blot analysis was performed according to the method previously reported [25,26]. Briefly, A549 and MCF-7 (1 × 10^5^ cells) were seeded into 6-wells plate and grown until 85–90% confluent. Then different concentrations (1.25, 2.5, 5, and 10 μM) of compound **1** were added. Cells were collected and lysed by radioimmunoprecipitation assay (RIPA) buffer. Lysates of total protein were separated by 12.5% sodium dodecyl sulfate-polyacrylamide gels and transferred to polyvinylidene difluoride (PVDF) membranes. After blocking, the membranes were incubated with anti-Bax, anti-Bcl-2 (Cell Signaling Inc., Danvers, MA, USA), anti-caspase-3, and anti-β-actin (GeneTex Inc., Irvine, CA, USA) primary antibodies at 4 °C overnight. Then each membrane was washed with Tris-buffered saline containing 0.1% Tween 20 (TBST) and incubated with horseradish peroxidase (HRP)-conjugated secondary antibodies at room temperature for 1 h while shaking. Finally, each membrane was developed using enhanced chemiluminescence (ECL) detection kit, and the images were visualized by ImageQuant LAS 4000 Mini biomolecular imager (GE Healthcare, Woburn, MA, USA). Band densities were quantified using ImageJ software (NIH, Bethesda, MD, USA).

#### 3.4.6. Statistical Analysis

Results were expressed as mean ± SEM, and comparisons were made applying Student’s *t*-test. A probability of 0.05 or less was deemed significant. The software Microsoft Excel 2016 was used for the statistical analysis.

## 4. Conclusions

Five new (including two with new carbon skeleton) and eleven known compounds have been isolated and identified from the aerial part of *E. fortunei*. Among these isolated compounds, compound **1** markedly induces apoptosis of A549 and MCF-7 cells through mitochondrial- and caspase-3-dependent pathways. The compound **1** belongs to hydrophobic molecule, so it can cross the cell membrane by passive diffusion, a nonselective process. During passive diffusion, compound **1** simply dissolves in the phospholipid bilayer, diffuses across it, and then dissolves in the aqueous solution at the other side of the membrane. During this process, no membrane proteins are involved and the direction of transport is determined simply by the relative concentrations of the molecule inside and outside of the cell. This suggests that eupatodibenzofuran A (**1**) is worth further investigation and may be expectantly developed as a candidate for the treatment or prevention of non-small-cell lung cancer and breast cancer.

## Data Availability

The data presented in this study are available in the main text and the Appendix A of this article.

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
