# Peer review of "Dibenzofuran, 4-Chromanone, Acetophenone, and Dithiecine Derivatives: Cytotoxic Constituents from Eupatorium fortunei"

_ijms, 2021, doi:10.3390/ijms22147448_

Round 1

Reviewer 1 Report

This manuscript reported five new compounds from Eupatorium fortune. Moreover, compound 1 exhibited potent antitumor activity. Article is well organized, while suffers from many necessary things at few instances. This reviewer rather likes to go for a revision before accepting this for publication in IJMS.

  1. In abstract, italics for Eupatorium fortune (line 16).
  2. Physicochemical properties could help the readers better understand the ability of compound 1 to cross the cell membrane. The data can be obtained by some open source websites or software.
  3. In line 54, the negative ion mode of mass spectrometric data is necessary.
  4. The flow cytometry and double dye experiment are necessary to illustrate the result of apoptosis.
  5. For possible target proteins, molecular docking is a good method.

Reviewer 2 Report

The manuscript reports the isolation and identification of 5 new natural compounds from the aerial parts of Eupatorium fortunei 

The isolation and charaterization of such new substances is well reported and properly assessed. Of interest is also the biomedical investigation of such compounds on cell assays which paves the way for further new applications

few typos were encountered durinf the manuscript reading and may be corrected during proofhead processing

Round 2

Reviewer 1 Report

I recommend acceptance.